# Activity of Antioxidant Enzymes and Their Association with Lipid Profile in Mexican People without Cardiovascular Disease: An Analysis of Interactions

**DOI:** 10.3390/ijerph15122687

**Published:** 2018-11-28

**Authors:** Susana Rivera-Mancía, Angélica Saraí Jiménez-Osorio, Omar Noel Medina-Campos, Eloísa Colín-Ramírez, Maite Vallejo, Ariadna Alcántara-Gaspar, Raúl Cartas-Rosado, Jesús Vargas-Barrón, José Pedraza-Chaverri

**Affiliations:** 1CONACYT—Instituto Nacional de Cardiología ‘Ignacio Chávez’, Juan Badiano 1, Sección XVI, Tlalpan, Ciudad de México 14080, Mexico; ecolinra@conacyt.mx (E.C.-R.); rcartasro@conacyt.mx (R.C.-R.); 2Laboratorio de Medicina Genómica del Hospital Regional Lic. Adolfo López Mateos, ISSSTE, Av. Universidad 1321, Florida, Álvaro Obregón, Ciudad de México 01030, Mexico; jimenez.osorio.as@gmail.com; 3Departamento Biología, Facultad de Química, Universidad Nacional Autónoma de México, Ciudad de México 04510, Mexico; omarnoelmedina@gmail.com (O.N.M.-C.); pedraza@unam.mx (J.P.-C.); 4Departamento de Investigación Sociomédica, Instituto Nacional de Cardiología ‘Ignacio Chávez’ Juan Badiano 1, Sección XVI, Tlalpan, Ciudad de México 14080, Mexico; maite_vallejo@yahoo.com.mx (M.V.); ari.wed@gmail.com (A.A.-G.); 5Dirección de Investigación, Instituto Nacional de Cardiología ‘Ignacio Chávez’ Juan Badiano 1, Sección XVI, Tlalpan, Ciudad de México 14080, Mexico; jesus.vargas@cardiologia.org.mx

**Keywords:** dyslipidemia, cardiovascular risk, antioxidant activity, paraoxonase, superoxide dismutase, ceruloplasmin, catalase, ferric-reducing ability of plasma

## Abstract

Dyslipidemia and oxidative stress are both considered to be factors involved in cardiovascular disease; however, the relationship between them has been little explored. In this work, we studied the association between the lipid profile and the activity of antioxidant enzymes such as paraoxonase-1 (PON1), superoxide dismutase 1 (SOD1), ceruloplasmin, and catalase, as well as total antioxidant capacity (the ferric-reducing ability of plasma (FRAP)), in 626 volunteers without cardiovascular disease. Their lipid profile was evaluated, and they were classified as having or not having high triglycerides (↑TG), high low-density cholesterol (↑LDLC), and low high-density cholesterol (↓HDLC), resulting in eight groups: Without dyslipidemia, ↑TG, ↑LDLC, ↓HDLC, ↑TG↑LDLC, ↑TG↓HDLC, ↑LDLC↓HDLC, and ↑TG↑LDLC↓HDLC. When comparisons by group were made, no significant differences in the activity of antioxidant enzymes were obtained. However, the linear regression analysis considering the potential interactions between ↑TG, ↑LDLC, and ↓HDLC suggested a triple interaction between the three lipid profile alterations on the activity of PON1 and a double interaction between ↑TG and ↑LDLC on ferroxidase-ceruloplasmin activity. The analysis presented in this work showed an association between the lipid profile and antioxidant-enzyme activity and highlighted the importance of considering the interactions between the components of a phenomenon instead of studying them individually. Longitudinal studies are needed to elucidate the nature of these associations.

## 1. Introduction

Dyslipidemia has been recognized as an important risk factor for atherosclerotic cardiovascular disease [1], and its prevalence has increased worldwide during the last years [2]. Likewise, it is known that oxidative stress is a fundamental process that contributes to the pathogenesis of cardiovascular disease [3]. Oxidative stress is the result of an imbalance between the production of reactive oxygen species and the ability of the antioxidant defense to deal with them, leading to oxidative damage to DNA, proteins, and lipids [3,4]. It has been hypothesized that oxidative stress could be the link between dyslipidemia and heart damage in rodents [5,6]. However, the extent to which dyslipidemia and the activity of antioxidant enzymes may be associated has been barely studied in humans [7,8], and there is more information in this respect from studies on animals [5]; for instance, reduced activity of antioxidant enzymes such as superoxide dismutase-1 (SOD1), catalase, and glutathione reductase has been observed in the liver, heart, and plasma of rats fed on a high-fat diet [5,6,9,10].

Of all the alterations in lipid profile and their relationship with antioxidant enzymes, the most studied is hypercholesterolemia [11]. In addition, the studies evaluating the activity of antioxidant enzymes in humans with dyslipidemia have been performed in patients with other underlying conditions [7,8] or under pharmacological treatment [7]—which could possibly have affected the antioxidant activity of the organism—and frequently by separating the groups into having or not having dyslipidemia, leaving aside the study of associations of the separate components of the lipid profile and their interactions. Therefore, it is not known whether the association between dyslipidemia and the activity of antioxidant enzymes exists in a context free of cardiovascular disease. For this reason, the aim of this work was to evaluate the association between the activity of the antioxidant enzymes SOD1, paraoxonase-1 (PON1), catalase, and ceruloplasmin; total antioxidant capacity (the ferric-reducing ability of plasma (FRAP)); and dyslipidemia, emphasizing the interactions between different altered components of the lipid profile, in volunteers without cardiovascular disease.

## 2. Materials and Methods

### 2.1. Volunteers

Volunteers were recruited from June to December 2015 as part of the Tlalpan 2020 cohort, a study aimed at evaluating the role of traditional and nontraditional factors on hypertension incidence in a population of Mexico City (for more details see Reference [12]). Volunteers were women and men aged 20 to 50 years old living in Mexico City and not suffering from hypertension or cardiovascular disease. People previously diagnosed with diabetes mellitus, dysthyroidism, cerebrovascular disease, ischemic cardiopathy, acute coronary syndrome, or cancer with an effect on survival, as well as pregnant women, people taking antihypertensive medication, or those with mental and cognitive disabilities were excluded. For the purpose of the present study, among recruited participants, those who had no complete data or were taking lipid-lowering medication were excluded, rendering a total of 626 volunteers (Figure 1). All participants signed their informed consent. The Tlalpan 2020 study followed the principles of the Declaration of Helsinki (revised in 2013) and was approved by the Institutional Bioethics Committee of the Instituto Nacional de Cardiología Ignacio Chávez (INCICh) (National Institute of Cardiology Ignacio Chavez) under number 13-802, whereas the substudy presented here was approved under number 15-947.

Information regarding smoking habit and alcohol consumption was recorded. Participants who reported to have smoked 100 or more cigarettes in their lifetime and, additionally, smoked daily or some days at the time of the survey, were classified as current smokers. Those who had smoked 100 cigarettes but did not smoke at the time of the survey were classified as ex-smokers, whereas those who had not smoked more than 100 cigarettes were considered nonsmokers [13]. People who at the time of the survey reported consuming alcohol, regardless of the frequency, were classified as current alcohol consumers.

To obtain physical activity information, the long version (7 days) of the International Physical Activity Questionnaire (IPAQ) was employed. This questionnaire was designed to assess physical activity in individuals 15–69 years old and evaluates four domains: work, home, transportation, and free time. The level of physical activity was categorized as low, moderate, or high, according to the criteria of the IPAQ working group [14].

Dietary intake was evaluated by means of a semiquantitative questionnaire, designed and validated in a Mexican population, which consisted of 116 questions about the frequency of consumption of food and beverages during the last year. Energy intake was estimated by using the computer program Sistema de Evaluación de Hábitos Nutricionales y Consumo de Nutrimentos (SNUT) (System of Evaluation of Nutritional Habits and Nutrient Consumption), which was developed by the National Institute of Public Health of Mexico [15].

As part of the routine evaluations, blood pressure, weight, and height were also measured in the volunteers of this study by using the procedures described in Reference [12]. Briefly, blood pressure was measured with a mercury sphygmomanometer (Riester, empire^®^ N, Jungingen, Germany) after the participants were seated for at least 10 min, and three measurements (with intervals of three minutes between them) were recorded and averaged. Height and weight were measured with the patient fasting, shoeless, and wearing a hospital gown in accordance with the International Society for the Advancement of Kinanthropometry (ISAK) [16].

### 2.2. Lipid Profile and Classification of Dyslipidemia

Venous blood samples were drawn from the participants after an overnight 12-h fasting period to determine total cholesterol (TC), triglycerides (TG), high-density lipoprotein cholesterol (HDLC) and low-density lipoprotein cholesterol (LDLC). Participants were classified as having or not having high TG (↑TG), high LDLC (↑LDLC), or low HDLC (↓HDLC) according to the Adult Treatment Panel III criteria [17] when triglycerides > 1.7 mmol/L, HDLC < 1.3 mmol/L for women and <1.0 mmol/L for men, and LDLC > 3.4 mmol/L. From the alterations in lipid profile (↑TG, ↑LDLC, and ↓HDLC), eight combinations were obtained: (1) without dyslipidemia, (2) only ↑TG, (3) only ↑LDLC, (4) only ↓HDLC, (5) ↑TG↑LDLC, (6) ↑TG↓HDLC, (7) ↑LDLC↓HDLC, and (8) ↑TG↑LDLC↓HDLC.

### 2.3. Activity of Antioxidant Enzymes and FRAP Assays

Aliquots of serum, plasma, and whole blood were kept at −70 °C until analysis.

#### 2.3.1. SOD1 Activity

The activity of SOD1 was determined in whole blood samples by the method of Oberley and Spitz [18], measuring the reduction of nitroblue tetrazolium (NBT) to formazan at 560 nm in a mixture reaction containing 0.122 mM ethylenediaminetetraacetic acid (EDTA), 30.6 μM NBT, 0.122 mM xanthine, 49 mM sodium carbonate, and 2.8 U/L of xanthine oxidase. One unit of SOD was defined as the amount of protein necessary to inhibit NBT reduction by 50%, and the enzyme activity was expressed as U/mg protein.

#### 2.3.2. Catalase Activity 

This parameter was determined by the method of Aebi [19], measuring the consumption of 30 mM hydrogen peroxide at 240 nm and expressing the enzyme activity as κ/mg protein.

#### 2.3.3. PON1 Activity 

Serum PON1 activity was measured by exposing the samples to paraoxon and determining the concentration of the resulting product, p-nitrophenol, at 412 nm. The activity of the enzyme was calculated by using the molar extinction coefficient of p-nitrophenol, 18,290 M^−1^ cm^−1^ [20], and it was reported as U/L.

#### 2.3.4. Ceruloplasmin Activity

The ferroxidase activity of ceruloplasmin was determined as previously reported [21,22,23], with slight modifications to read the final reaction in a microplate reader (SPECTROstar nano, BMG Labtech). Briefly, 100 µL of plasma samples (diluted 1:25 in acetate buffer 0.3 M, pH 6) were placed in microtubes, and their temperature was stabilized at 30 °C for 10 min. Later, 100 µL of 10 mM ammonium iron (II) sulfate were added to the samples, and they were incubated at 30 °C for 5 min. The reaction was stopped by adding 200 µL of 1.25 M perchloric acid. Microtubes were centrifuged at 10,000× *g* for 3 min. One-hundred µL of the supernatant were transferred to a microplate, and 100 µL of 0.5 M sodium thiocyanate were added to each well. The plate was immediately shaken, and the colorful complex was read at 450 nm. Absorbance was interpolated in a calibration curve constructed with a standard solution of ammonium iron (III) sulfate. Results were expressed as U/mL.

#### 2.3.5. FRAP Assay 

Antioxidant total capacity was determined by using a working solution containing 0.833 mM 2,4,6-Tris(2-pyridyl)-s-triazine (TPTZ) and 1.66 mM FeCl_3_ in acetate buffer (300 mM pH 3.6). Thirty µL of sample were mixed with 300 µL of working solution and incubated for 15 min. The optical density was read at 593 nm, and these values were interpolated in a Trolox [(±)-6-Hydroxy-2,5,7,8-tetramethylchromane-2-carboxylic acid] standard curve [24]. The results were expressed as micromoles of Trolox equivalents.

### 2.4. Statistical Analysis

The general characteristics of participants were represented as mean ± standard deviation or median with interquartile range, in the case of continuous variables. For categorical variables, data were displayed as frequencies and percentages. To compare continuous variables between genders, Student’s *t*-test (when homogeneity of variances was found) or the Mann–Whitney U test (when there was no homogeneity of variances) were performed, whereas the chi-squared test was carried out for those that were categorical.

Mean ± standard deviation for the activity of PON1, SOD1, catalase, and ceruloplasmin, as well as the FRAP, were obtained for each type of dyslipidemia that resulted from the combinations of the alterations of the lipid profile (↑TG, ↑LDLC, and ↓HDLC) for descriptive purposes, and they were compared using ANOVA followed by the Bonferroni post hoc test. A supplementary bivariate analysis for the associations between the activity of the antioxidant parameters and factors associated with dyslipidemia or antioxidant activity is provided in Appendix A, whereas the bivariate correlations between all of these antioxidant parameters are reported in the Appendix A.

The main analysis for this study consisted of multiple linear regression models for the associations between the activity of each of the enzymes mentioned in Section 2.3 and the combinations of the alterations in the lipid profile, considering relevant covariates such as age, gender, alcohol consumption, caloric intake, physical activity, and tobacco smoking: Dummy variables were created for these last two variables due to them having three categories each. The terms for interactions between high-TG, high-LDLC, and low-HDLC were included in the linear regression models: ↑TG × ↑LDLC, ↑TG × ↓HDLC, ↑LDLC × ↓HDLC, and ↑TG × ↑LDLC × ↓HDLC, giving place to a linear regression model of the type *Y* = α + β × *X*_1_ + β_2_ × *X*_2_ + β_3_ × *X*_3_ + β_12_ × *X*_1_ × *X*_2_ + β_13_ × *X*_1_ × *X*_3_ + β_23_ × *X*_2_ × *X*_3_ + β_123_ × *X*_1_ × *X*_2_ × *X*_3_ + ε. The assumptions of normality of residuals, homoscedasticity, and non-multicollinearity were verified in each case, and when they were not met, log 10 transformations were made. The standardized β coefficients for each lipid profile-related term of the linear models were obtained for the unadjusted models and for the models including all the covariates.

## 3. Results

### 3.1. General Characteristics of Participants

The general characteristics of the volunteers are shown in Table 1. Of the 626 participants included in this study, 63.1% were women, who displayed a significantly better lipid profile compared to men except for total cholesterol (*p* = 0.225), as well as lower blood pressure and weight. Significant differences regarding alcohol use and tobacco smoking were also found, both being more frequent in men than in women. The frequency of men in the category of high physical activity was significantly higher than women (Table 1).

The prevalence of each combination that resulted from the different alterations of the lipid profile was estimated for men and women, and we found that only near one-third of the sample was free of dyslipidemia. The most prevalent combination of dyslipidemia for women was “only ↓HDLC”, whereas for men “↑TG↑LDLC” and “↑TG↓HDLC” were the most frequent combinations (Table 1). The combinations with the lowest prevalence were “only ↑TG” for women and “↑LDLC↓HDLC” for men.

Mean ± SD were obtained for every enzyme activity and FRAP (Table 2). When comparisons between groups were made, significant differences were only observed for FRAP between the ↑TG↓HDLC and other three groups: without dyslipidemia, ↑LDLC, and ↓HDLC. For the activity of antioxidant enzymes, no significant differences were obtained, but there was a trend to lower paraoxonase activity for groups with low HDLC (except for the ↑TG ↑LDLC ↓HDLC group), whereas for ceruloplasmin activity the ↑TG and the ↑LDLC groups tended to be the ones with the highest activity. Those groups combining both of these alterations tended to have the lowest activity for both ceruloplasmin and SOD1 (Table 2).

Results, as seen in Appendix A, showed a lower FRAP for nonsmokers, a higher activity of SOD1 in women compared to men, and the inverse behavior for catalase and FRAP. They also showed a slight negative correlation between FRAP and age and slight positive correlations for PON1 and FRAP with caloric intake, whereas this last parameter negatively correlated with SOD1 activity.

### 3.2. Linear Regression Models for the Association between Dyslipidemia and Antioxidant Enzymes

Linear regression models were constructed with the activity of each enzymes or FRAP as dependent variables and the alterations of the lipid profile components (as dichotomic variables, yes or no) as the independent ones. Catalase activity and FRAP were introduced into the models as log(10)-transformed variables given that the respective models did not meet the assumption of normality of residuals. Table 3 shows the liner regression models considering the alterations of the components of the lipid profile individually (↑TG, ↑LDLC, and ↓HDLC), whereas Table 4 and Appendix A present models including their interactions (↑TG × ↑LDLC, ↑TG × ↓HDLC, ↑LDLC × ↓HDLC, and ↑TG × ↑LDLC × ↓HDLC).

When the combinations of altered lipid profile were introduced in the models as main effects only (individual factors only), a significant negative association was observed between ↓HDLC and PON1 activity (β = −27.12, 95% CI: −49.75, −4.50, *p* = 0.019), and a trend toward a positive association between ↑TG and the activity of PON1 was also found (β = 23.23, 95% CI: −1.03, 47.50, *p* = 0.061, for the adjusted model), whereas no significant association was obtained for the other enzymes. Log FRAP was positively associated with ↑TG (β = 0.03, 95% CI: 0.00, 0.06, *p* < 0.041) and ↓HDLC (β = 0.03, 95% CI: 0.01, 0.06, *p* = 0.018, for the adjusted model).

In the models considering interactions, a triple significant interaction between ↑TG, ↑LDLC, and ↓HDLC was observed for the activity of PON1, with a positive β coefficient in the unadjusted model (β = 105.81, 95% CI: 3.37, 208.24, *p* = 0.043), although the *p*-value changed to 0.072 when all the covariates (age, gender, smoking and alcohol use, physical activity, and caloric intake) were included in the model. However, none of the covariates were statistically significant, either individually or as a whole. It is worth mentioning that an additional analysis without outliers for the activity of PON1 was performed (seven outliers were detected), leading to significant associations for both the unadjusted and adjusted models (β (unadjusted model) = 131.65, 95% CI: 33.68, 229.62, *p* = 0.009; β (adjusted model) = 121.28, 95% CI: 22.84, 219.72, *p* = 0.016).

A triple interaction means that a double interaction exists, but it depends on the status of a third variable. To detect which were the double interactions and what was the third variable, the analysis was split first into ↑LDLC and normal LDLC to test the ↑TG-↓HDLC double interaction, then into ↑TG and normal TG to test the ↑LDLC-↓HDLC double interaction, and finally into ↓HDLC and normal HDLC to test the ↑LDLC-↑TG double interaction. Results from this subanalysis are presented in Table 5. We found an interaction between ↑LDLC and ↓HDLC (*p* = 0.031) when TGs were high (the statistical significance of this interaction was attenuated when all the covariates were included in the model (*p* = 0.074), although none of them had a significant effect), and an interaction between ↑LDLC and ↑TG was observed in the presence of ↓HDLC. Then, when levels of TG were high, the combination of ↓HDLC and ↑LDLC gave rise to a level of activity of PON1 that was higher than expected. Additionally, when the status of HDLC was fixed as low in the analysis, we observed that the combination of ↑TG with ↑LDLC was associated with higher activity of PON1 (*p* = 0.007 for the interaction without covariates, and *p* = 0.011 when the model was adjusted for covariates).

In the case of ceruloplasmin, a significant interaction was observed between ↑TG and ↑LDLC with a negative β coefficient (standardized β coefficient = −0.254, *p* = 0.020). The statistical significance was maintained regardless of the covariates included in the model (standardized β coefficient = −0.265, *p* = 0.016).

For the other antioxidant enzymes, catalase and SOD1, and for FRAP, none of the lipid profile-associated terms showed a statistically significant association. Nevertheless, for SOD1, gender was a statistically significant covariate (β = −0.68, CI 95% −1.21, −0.15, *p* = 0.012). For catalase activity (log catalase for the purpose of the analysis), only physical activity (β for moderate physical activity = −0.07, CI 95% −0.14, −0.01, *p* = 0.034; β for high physical activity = −0.08, CI 95% −0.15, −0.02, *p* = 0.014) was statistically significant in the model adjusted by all the covariates. In the case of FRAP (log FRAP for the purpose of the analysis), age (β = −0.002, CI 95% −0.004, −0.001, *p* = 0.001), gender (β = 0.10, CI 95% 0.07, 0.12, *p* < 0.001), and physical activity (β for high physical activity = 0.05, CI 95% 0.01, 0.09, *p* = 0.017) had a significant effect.

## 4. Discussion

To our knowledge, this is the first time that interactions between altered components of the lipid profile have been evaluated regarding their associations with the activity of antioxidant enzymes, highlighting the relevance of considering not only the individual factors of a phenomenon, but the interactions between them. The strategy of studying interactions has the advantage of allowing observations on how a factor may influence the effect of another, something that is not possible when each factor is evaluated separately [25]. We found that combinations of altered lipid profile were associated with differences in the activity of antioxidant enzymes, particularly PON1 and ceruloplasmin. We will discuss the findings on PON1 activity more widely to describe the triple interaction we found.

PON1 is an enzyme with antioxidant and anti-atherogenic properties, whose reduced activity has been linked to heart disease in humans [26]. One of its functions is to protect LDL and HDL from oxidation, thus avoiding the increase of oxidized LDL [27]. Although the main analysis was based on evaluating interactions among the components of the lipid profile, a significant association between ↓HDLC and reduced activity of PON1 was found in the analysis of individual factors (Table 2), and the activity of the enzyme also tended to be lower in those combinations of altered lipid profile that included a low level of HDLC (Table 2), except for the ↑TG × ↑LDLC × ↓HDLC combination. The association between ↓HDLC and reduced PON1 activity was in agreement with the fact that PON1, after being synthesized by the liver, is physically associated with HDL in the plasma [28].

The analysis including the interaction terms showed a triple interaction between ↑TG, ↑LDLC, and ↓HDLC for the activity of PON1, which means that the combination of the three factors was different from the sum of the separate effects. One would expect that when HDLC was low, the activity of PON1 should be low. However, the subanalysis of the triple interaction (Table 5) showed that the interaction between ↑LDLC and ↑TG that was not significant (Table 4 (*p* = 0.887)) when the effect of ↓HDLC was not considered, became significant (*p* = 0.011) only when HDLC was low. Thus, under those very specific conditions, the combination of ↑LDLC with ↑TG was associated with a relatively high level of activity of PON1 that was not in accordance with the presence of ↓HDLC.

It is possible that the triple interaction was the result of some influence of ↑LDLC on the activity of the enzyme only when ↓HDLC and ↑TG coexisted. Regarding this possible influence of LDLC on the activity of PON1, Rozek et al. [26] reported that LDLC positively correlated with the activity of PON1 in individuals without carotid artery disease (CAD), and such an association was not observed in individuals with CAD or those without CAD but taking lipid-lowering medication. On the contrary, in a study performed in Chinese men, PON1 activity was negatively correlated with LDLC [29]. Interestingly, Rozek et al. [26] also reported a positive correlation between PON1 activity and TG and very low-density lipoproteins (which are intimately linked to TG levels) only in the group with CAD, and no association with HDL3 particles in patients with CAD despite the physical association that exists between PON1 and those particles. From this evidence, it can be suggested that the high variability of PON1 activity and the inconsistencies in studies of the association between PON1 activity and cardiovascular diseases [30,31] cannot only be due to pathophysiological conditions and ethnicity [32], but also to the variety of alterations of the lipid profile.

Although we cannot propose a molecular mechanism to explain the association between lipid profile and PON1 activity from our data, it is important to mention that there are studies in animals and humans that have provided information about the regulation of PON1 expression by lipoproteins, indicating that sterol regulatory binding protein-2 (SREBP-2) could regulate the transcription of the PON1 gene [31]. It has also been suggested that SREBP-2 effects are dependent on LDL levels [26], which could partially explain the potential influence of LDLC on the interaction we found.

Regarding ceruloplasmin, this is a copper-dependent enzyme with ferroxidase and antioxidant activity [33] whose high concentration is considered as an independent predictor of cardiovascular events [34]. Due to dyslipidemia also being considered a risk factor for cardiovascular diseases [1], we expected to find increased ceruloplasmin activity associated with dyslipidemia. However, we found that whereas ↑TG and ↑LDLC separately tended to increase the activity of the enzyme, when both of these alterations were together there was no sum of effects. On the contrary, a change in the direction of the association between ↑TG and ↑LDLC with the activity of the enzyme was found. Here, it is worth mentioning that the majority of studies of the association between ceruloplasmin and cardiovascular risk have evaluated the concentration of the enzyme instead of its activity [34,35]. However, in a study by Buyukhatipoglu et al. [7], they found that CAD dyslipidemic patients under treatment with statins, which had lower levels of LDLC, displayed higher ceruloplasmin activity than those under statin medication. This relationship is, in a certain way, similar to our findings, as the group with the lowest LDLC levels (the highest statin dose) was the one with the highest activity of the enzyme: However, there were no significant differences in TGs between the groups treated with statins and the control group. More in accordance with our results, a high-cholesterol diet increased serum LDLC and triglycerides in rabbits under treatment with a high cholesterol diet showed increased serum LDLC and TGs, along with lower ceruloplasmin activity compared to the control group [36]; the observed differences were attributed to an affection of the transport of divalent metals, as the supplementation with minerals such as copper and zinc partially or fully restored the activity of the enzyme. Thus, it is possible that a high-cholesterol diet or the presence of ↑LDLC and ↑TG in serum disturbs the homeostasis of metals that affect the activity of ceruloplasmin.

SOD1 activity only displayed a trend of decreasing when ↑TG and ↑LDLC coexisted and increasing when ↑LDLC was combined with ↓HDLC, but the associations did not reach statistical significance. Some studies have found that reduced SOD1 activity was associated with the presence of dyslipidemia [37], whereas others have not [38]. However, the involvement of SOD1 in lipid metabolism has been revealed in studies in animals, mainly regarding the secretion of lipoproteins [39]. One possible explanation for the discrepancies about the association between SOD1 activity and dyslipidemia could be genetic differences, as polymorphisms of SOD1 have been linked to differences in fat metabolism, for instance the −251 A > G SOD1 (rs2070424) polymorphism [40]. We found no reports in the literature in relation to differences in SOD1 activity between genders. However, we found here that women had higher activity of this enzyme than men (Appendix A), and from the linear regression analysis we observed that being male was associated with lower activity of SOD1 0.686 U/mg, whereas studies evaluating the expression of the enzyme have found higher expression in men [41], although it should be kept in mind that gene expression is not necessarily correlated with protein amount. Thus, the relevance of this finding and its implications need to be studied in depth.

For catalase, no statistically significant association with dyslipidemia was found in our study. As for SOD1, results are contradictory, as both reduced [37] and increased [42] activity of the enzyme has been found in patients with dyslipidemia. Again, these discrepancies may be attributed to the genetic background of the studied population, due to some catalase polymorphisms having been associated with lipid levels. For instance, the CAT-262CC polymorphism was associated with higher triglyceride levels in a Greek population [43], whereas the individuals possessing the −21 A > T CAT (rs7943316) mutation in a Mexican population responded better to a dietary intervention compared to those individuals without the mutation, improving their lipid profile [40].

The last parameter we evaluated was FRAP, which reflects the antioxidant capacity given by the non-enzymatic components of plasma [24]. When individual components were analyzed, a significant association was observed for ↑TG and for ↓HDLC, and no effect of interactions was observed, meaning that the components of the lipid profile did not affect each other regarding FRAP. Hypertriglyceridemia was also found to be associated with increased FRAP in the study by Miri et al. [42]; in this respect, it must be recalled that uric acid importantly contributes to the antioxidant power that is evaluated by this technique [24] and that uric acid is usually elevated in dyslipidemia, positively correlated with TG, and negatively correlated with HDLC [44]. FRAP was positively associated with high physical activity and with being male, and negatively associated with age, as reported in other studies [45,46,47] and in agreement with Appendix A.

From our study, we also obtained some data regarding the prevalence of different types of dyslipidemia. We found that only one-third of our sample was free of dyslipidemia and that the prevalence of altered lipid profile was very different between men and women, with the ↑TG-↑LDLC combination the most prevalent among men and ↓HDLC among women. A study by Frank et al. [32] showed differences in the lipid profile by gender and ethnicity: For instance, ↑LDLC was more common in Filipinos than in other races and more common in men than in women. On the other hand, ↓HDLC was more common in Mexicans and Asian Indians but, whereas its prevalence was higher in women in some populations (e.g., Asian Indians, Koreans, Mexicans, and black/Africans), it was higher in men for others (e.g., non-Hispanic whites, Chinese, and Japanese), hence the importance of studying data involving dyslipidemia for every specific population. In addition, we reported some associations between the antioxidant parameters measured in our study (Appendix A) that showed that most of them were interrelated, with SOD1 activity the only one that was correlated with all the other parameters. These associations need to be further studied.

It is worth mentioning that some statistical considerations on this work could be made. We obtained significant interactions in our factorial models, which in practice are very unlikely to occur [25,48,49]. Thus, if they are observed, there is a good chance that they are true interactions. However, there is now the justifiable trend to carry out adjustments for multiplicity in linear regression models, with the purpose of avoiding false positives. For this reason, we are reporting the calculation of *p*-critical values by using the procedure of Benjamini and Hochberg [50] as Appendix A, although it should be taken into account that this type of procedure is applied when independent comparisons or hypothesis are being tested [50,51], and in our case the variables (↓HDLC, ↑LDLC, and ↑TG) were conditioned to each other because of the interaction terms. If the calculated *p*-critical values were used for our hypothesis testing, then only the effect of ↑TG on FRAP in the model that only contained main effects would be statistically significant. Due to McDonald [51] recommending reporting both the analysis adjusted by multiplicity and the unadjusted one, to avoid the loss of potentially relevant biological findings, they both were reported in this manuscript.

One main limitation of this study is that we could not ascertain that the alterations we observed concerning the activity of ceruloplasmin and PON1 reflected the status of these enzymes in specific organs, or a causality relationship between dyslipidemia and these alterations, due to the cross-sectional design of this work. It is possible that even the relationship between antioxidant enzymes and dyslipidemia had an inverse directionality to that studied here. For instance, the status of antioxidant enzymes might have affected the status of serum lipids. To be sure about a causal relationship between dyslipidemia and modifications of the activity of antioxidant enzymes and their consequences, a longitudinal study could shed more light in this regard. In addition, we identified two potential biases of this study: (1) The participants did not constitute a random sample, as they voluntarily attended to be evaluated at the INCICh, limiting the generalizability of the results, and (2) data on physical activity, dietary intake, alcohol consumption, and smoking habits were self-reported by the volunteers, so they were subject to memory bias.

## 5. Conclusions

Lipid profile alterations were associated with differences in the activity of antioxidant enzymes in our sample. Particularly, the interaction between ↓HDLC, ↑TG, and ↑LDLC was associated with a relatively high activity of PON1, regardless of the low levels of HDLC, whereas the interaction between ↑TG and ↑LDLC was associated with an activity of ceruloplasmin that was opposed to that found for the separate components (↑TG and ↑LDLC). The relevance of these associations lies in the fact that dyslipidemia and disrupted antioxidant activity are related to cardiovascular risk and that they can even be found in individuals that are free of cardiovascular diseases, as in this case. We consider that the observations presented here deserve further research from the mechanistic point of view to find the potential influence of the lipid profile on the activity of antioxidant enzymes or vice versa.

## Figures and Tables

**Figure 1 ijerph-15-02687-f001:**
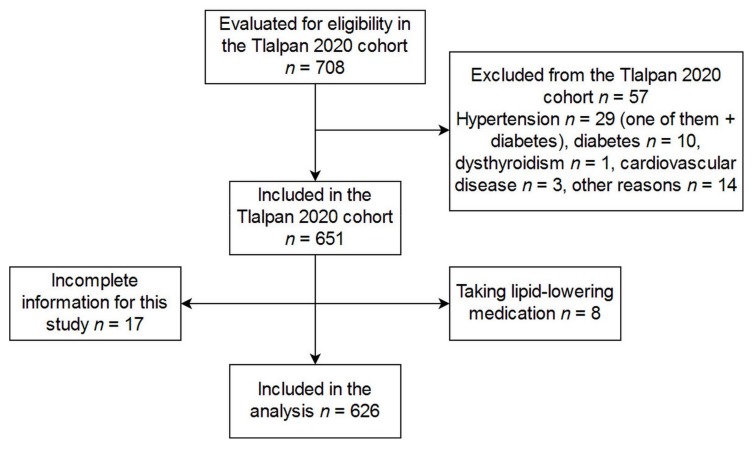
Flow diagram of participants. This study included volunteers from the Tlalpan 2020 cohort. In addition to the criteria of the cohort, those participants taking lipid-lowering medication were excluded from this study.

**Table 1 ijerph-15-02687-t001:** General characteristics of participants.

Characteristic		All (*n* = 626)	Women (*n* = 395)	Men (*n* = 231)	*p*-Value
Age (years)		37.12 ± 8.97	37.67 ± 8.88	36.18 ± 9.06	0.046
Systolic blood pressure (mm Hg)		106.66 ± 10.65	104.33 ± 10.29	110.64 ± 10.07	<0.001
Diastolic blood pressure (mm Hg)		71.23 ± 8.14	69.64 ± 7.89	73.94 ± 7.86	<0.001
Weight (kg)		71.13 ± 14.68	66.31 ± 12.81	79.36 ± 14.03	<0.001
Body mass index (kg/m^2^)		26.94 ± 4.56	26.83 ± 4.75	27.14 ± 4.25	0.411
Triglycerides (mmol/L)		1.36 (0.96, 2.00)	1.24 (0.89, 1.71)	1.75 (1.15, 2.32)	<0.001
Total cholesterol (mmol/L)		4.85 ± 0.92	4.82 ± 0.91	4.91 ± 0.95	0.225
LDL-cholesterol (mmol/L)		3.14 ± 0.79	3.08 ± 0.79	3.24 ± 0.79	0.012
HDL-cholesterol (mmol/L)		1.22 (1.03, 1.46)	1.32 (1.11, 1.56)	1.10 (0.93, 1.26)	<0.001
Caloric intake (kcal)		2233.93 (1503.86, 2738.71)	2063.82 (1455.35, 2511.08)	2606.35 (2095.66, 3146.86)	<0.001
Tobacco smoking [*n* (%)]	Never	395 (63.1)	281 (71.1)	114 (49.4)	<0.001
	Former	105 (16.8)	47 (11.9)	58 (25.1)	
	Current	126 (20.1)	67 (17.0)	59 (25.5)	
Alcohol use [*n* (%)]	Yes	450 (71.9)	273 (69.1)	177 (76.6)	0.044
	No	176 (28.1)	122 (30.9)	54 (23.4)	
Physical activity [*n* (%)]	Low	64 (10.2)	38 (9.6)	26 (11.3)	0.003
	Medium	263 (42.0)	186 (47.1)	77 (33.3)	
	High	299 (47.8)	171 (43.3)	128 (55.4)	
Combination of alterations of lipid profile [*n* (%)]	Without dyslipidemia	197 (31.5)	132 (33.4)	65 (28.1)	<0.001
	Only ↑TG	24 (3.8)	6 (1.5)	18 (7.8)	
	Only ↑LDLC	75 (12.0)	55 (13.9)	20 (8.7)	
	Only ↓HDLC	101 (16.1)	81 (20.5)	20 (8.7)	
	↑TG↑LDLC	58 (9.3)	17 (4.3)	41 (17.7)	
	↑TG↓HDLC	82 (13.1)	44 (11.1)	38 (16.5)	
	↑LDLC↓HDLC	28 (4.5)	25 (6.3)	3 (1.3)	
	↑TG↑LDLC↓HDLC	61 (9.7)	35 (8.9)	26 (11.3)	

Results for continuous variables are expressed as means ± SD or median (interquartile range), and for categorical variables they are expressed as frequencies and percentages. High triglycerides: ↑TG; high low-density lipoprotein cholesterol: ↑LDLC; low high-density lipoprotein cholesterol: ↓HDLC.

**Table 2 ijerph-15-02687-t002:** Descriptive data of the antioxidant enzymes activity and the ferric-reducing ability of plasma (FRAP).

Parameter	Without Dyslipidemia	↑TG	↑LDLC	↓HDLC	↑TG ↑LDLC	↑TG ↓HDLC	↑LDLC ↓HDLC	↑TG ↑LDLC ↓HDLC
Paraoxonase-1 activity (U/mL)	278.9 ± 131.0	321.6 ± 102.7	266.9 ± 142.8	257.4 ± 127.9	298.9 ± 165.1	243.7 ± 117.6	228.4 ± 122.3	309.6 ± 145.6
Ceruloplasmin activity (U/mL)	2.28 ± 0.31	2.37 ± 0.41	2.35 ± 0.36	2.26 ± 0.26	2.24 ± 0.34	2.30 ± 0.36	2.34 ± 0.32	2.28 ± 0.31
Superoxide dismutase-1 activity (U/mg)	11.1 ± 2.8	11.2 ± 2.9	11.3 ± 3.0	11.0 ± 2.7	10.5 ± 2.7	11.0 ± 3.2	11.9 ± 3.3	10.8 ± 2.4
Catalase activity (κ/mg)	0.34 ± 0.18	0.34 ± 0.15	0.33 ± 0.20	0.32 ± 0.18	0.35 ± 0.16	0.33 ± 0.18	0.34 ± 0.19	0.31 ± 0.15
FRAP (micromoles of Trolox equivalents)	953.1 ± 352.0 ^a^	1048.6 ± 383.9	964.2 ± 332.3 ^b^	964.2 ± 328.9 ^c^	1125.3 ± 364.6	1162.9 ± 526.1 ^a,b,c^	1021.9 ± 315.0	1069.4 ± 380.1

Data are displayed as mean ± SD and are presented only for descriptive purposes. Those groups marked with the same superscript were statistically different in the ANOVA followed by the Bonferroni post hoc test. ^a^
*p* = 0.001, ^b^
*p* = 0.029, ^c^
*p* = 0.012.

**Table 3 ijerph-15-02687-t003:** Results from the linear regression models considering the main effects of the components of lipid profile and their association with the activity of antioxidant enzymes and FRAP.

Parameter	Term in the Equation	Unadjusted β Coefficient (95% CI)	Unadjusted Standardized β Coefficient	*p*-Value	Adjusted β Coefficient (95% CI)	Adjusted Standardized β Coefficient	*p*-Value
Paraoxonase-1 activity	↑TG	23.23 (−1.03, 47.50)	0.082	0.061	22.59 (−3.33, 48.50)	0.080	0.087
	↑LDLC	5.17 (−18.05, 28.39)	0.018	0.662	5.79 (−17.97, 29.54)	0.020	0.478
	↓HDLC	−27.12 (−49.75, −4.50)	−0.099	0.019	−29.31 (−52.48, −6.14)	−0.107	0.013
Superoxide dismutase-1 activity	↑TG	−0.38 (−0.89, 0.13)	−0.065	0.143	−0.11 (−0.65, 0.44)	−0.018	0.705
	↑LDLC	0.06 (−0.43, 0.55)	0.010	0.818	0.01 (−0.51, 0.50)	−0.001	0.983
	↓HDLC	0.11 (−0.37, 0.58)	0.019	0.662	−0.05 (−0.54, 0.43)	−0.010	0.829
Log catalase activity	↑TG	0.02 (−0.03, 0.06)	0.035	0.428	−0.01 (−0.05, 0.04)	−0.010	0.833
	↑LDLC	0.00 (−0.04, 0.04)	0.002	0.964	0.00 (−0.04, 0.04)	−0.002	0.964
	↓HDLC	−0.03 (−0.07, 0.01)	−0.070	0.100	−0.03 (−0.07, 0.01)	−0.061	0.162
Ceruloplasmin activity	↑TG	−0.01 (−0.07, 0.05)	−0.019	0.664	−0.02 (−0.09, 0.04)	−0.033	0.486
	↑LDLC	0.02 (−0.04, 0.07)	0.024	0.574	0.01 (−0.04, 0.07)	0.021	0.631
	↓HDLC	−0.01 (−0.07, 0.04)	−0.017	0.696	−0.01 (−0.06, 0.05)	−0.008	0.860
Log ferric-reducing ability of plasma	↑TG	0.06 (0.03, 0.08)	0.175	<0.001	0.03 (0.00, 0.06)	0.089	0.041
	↑LDLC	0.01 (−0.02, 0.03)	0.023	0.575	0.02 (−0.01, 0.05)	0.060	0.131
	↓HDLC	0.01 (−0.02, 0.04)	0.029	0.490	0.03 (0.01, 0.06)	0.096	0.018

Shaded cells are those where *p* < 0.05. The adjusted β coefficient resulted from the linear models adjusted for all the covariates (age, gender, tobacco smoking, alcohol use, physical activity, and caloric intake).

**Table 4 ijerph-15-02687-t004:** Results from the linear regression models considering the main effects of the components of lipid profile and their double and triple interactions associated with the activity of paraoxonase-1 (PON1) and ceruloplasmin.

Parameter	Term in the Equation	Unadjusted β Coefficient (95% CI)	Unadjusted Standardized β Coefficient	*p*-Value	Adjusted β Coefficient (95% CI)	Adjusted Standardized β Coefficient	*p*-Value
Paraoxonase-1 activity	↑TG	42.28 (−14.62, 99.17)	0.150	0.145	36.61 (−21.70, 94.93)	0.130	0.218
	↑LDLC	−12.41 (−48.12, 23.30)	−0.044	0.495	−10.66 (−47.13, 25.81)	−0.038	0.566
	↓HDLC	−21.44 (−53.64, 10.77)	−0.079	0.192	23.77 (−56.17, 8.63)	−0.087	0.150
	↑TG × ↑LDLC	−10.28 (−83.46, 62.89)	−0.030	0.783	−5.33 (−78.85, 68.19)	−0.015	0.887
	↑TG × ↓HDLC	−56.42 (−125.46, 12.63)	−0.175	0.109	−48.46 (−117.87, 20.96)	−0.151	0.171
	↑LDLC × ↓HDLC	−17.08 (−83.66, 49.51)	−0.044	0.615	−17.26 (−83.91, 49.39)	−0.045	0.611
	↑TG × ↑LDLC × ↓HDLC	105.81 (3.37, 208.24)	0.232	0.043	94.32 (−8.39, 197.03)	0.207	0.072
Ceruloplasmin activity	↑TG	0.09 (−0.044, 0.23)	0.137	0.186	0.09 (−0.05, 0.23)	0.130	0.225
	↑LDLC	0.07 (−0.016, 0.16)	0.104	0.111	0.07 (−0.02, 0.16)	0.109	0.103
	↓HDLC	−0.03 (−0.10, 0.05)	−0.038	0.527	−0.02 (−0.10, 0.06)	−0.029	0.632
	↑TG × ↑LDLC	−0.21 (−0.38, −0.03)	−0.254	0.020	−0.22 (−0.39, −0.04)	−0.265	0.016
	↑TG × ↓HDLC	−0.05 (−0.22, 0.11)	−0.069	0.530	−0.05 (−0.22, 0.12)	−0.068	0.542
	↑LDLC × ↓HDLC	0.016 (−0.14, 0.18)	0.081	0.848	0.016 (−0.15, 0.18)	0.017	0.849
	↑TG × ↑LDLC × ↓HDLC	0.10 (−0.15, 0.34)	0.088	0.448	0.093 (−0.15, 0.34)	0.126	0.459

Shaded cells are those where *p* < 0.05. The adjusted β coefficient resulted from the linear models adjusted for all the covariates (age, gender, tobacco smoking, alcohol use, physical activity, and caloric intake).

**Table 5 ijerph-15-02687-t005:** Subanalysis of the triple interaction between ↑ TG, ↑ LDLC, and ↓ HDLC and PON1 activity.

Double Interaction Tested	Fixed Condition	Unadjusted β Coefficient (CI 95%)	Unadjusted Standardized β Coefficient	*p*-Value	Adjusted β Coefficient (CI 95%)	Adjusted Standardized β Coefficient	*p*-Value
↑TG × ↓HDLC	Normal LDLC	−56.42 (−121.38, 8.55)	−0.179	0.089	−52.79 (−118.15, 12.57)	−0.167	0.113
	↑LDLC	49.39 (−34.28, 133.06)	0.148	0.246	52.14 (−31.17, 135.45)	0.156	0.219
↑LDLC × ↓HDLC	Normal TG	−17.08 (−72.63, 48.48)	−0.033	0.609	−16.96 (−82.95, 49.04)	−0.033	0.614
	↑TG	88.73 (8.34, 169.12)	0.281	0.031	74.55 (−7.30, 156.40)	0.236	0.074
↑TG × ↑LDLC	Normal HDLC	−10.28 (−85.71, 65.15)	−0.028	0.789	−3.29 (−79.19, 72.61)	−0.009	0.932
	↓HDLC	95.53 (26.50, 164.55)	0.305	0.007	91.59 (21.18, 162.00)	0.293	0.011

To know which combination of factors gave rise to the triple interaction between components of the lipid profile for PON1 activity, interactions between two components (double interactions in first column of the table) were evaluated with a fixed condition for a third component (second column of the table). Shaded cells are those where *p* < 0.05.

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
