# Peer review of "Activity of Antioxidant Enzymes and Their Association with Lipid Profile in Mexican People without Cardiovascular Disease: An Analysis of Interactions"

_ijerph, 2018, doi:10.3390/ijerph15122687_

Round 1

Reviewer 1 Report

This manuscript is an analysis of Baseline characteristics in a propective study in Mexican subjects the Tlalpan 2010 Study.

The design paper in BMJ open does not describe this substudy of anti-oxidant activities acccording to risk factors.

Six antioxidant actitivities have been measured in 626 men and women. The major issue is the focus given to the manuscript with multiple statistical analyses without any adjustment for multiplicity. 15 comparisons in table 2, 35 in table 3 of which  of 3 are statisticant at the p<0.05 level, which could represent simply chance findings. This is threatening the full construct based on double and triple interactions.

The proposal is to fully reorganize the manuscript to describe the measured antoxidant activities and their relationsghips according to lipid levels, their defined subgroups and the risk factors, precisely documented in this prospective cohort. That would largely increase  the merit and the interest to readers of the study for readers.

The title should state that the selected population does not have cardiovascular disease and types of dyslipidemia replaced by lipid profile.

Abstract line 30 state separately; line 33 state ceruleoplasmin;  rewrite the main finding and in the conclusion perplace mechanistic by prospective or longitudinal studies.

Introduction

rewrite line 59-60 has this manuscript does not report events but their risk

2.1 subjects and risk factors are fine; consider however to add blood pressure anf weight figures

2.2 please move here  the classification given line 171-174

2.3  antioxidant activities explain how SOD1  and catalase activities were reported in U/mg or ?kappa/mg protein

2.4 statistical analysis. this is the main isuue with proposal to abandon most of statistical analysis except those associated with the lasrt 2 rows of  table 4 on paraoxonase.. line 146 what is meant by considering relevant; line 152 which log transformation were done

Results 

Table 1 please detete the stars which are not necessary

Table 2 should be table 5 with the same order for parameters than the one in the method section

Table 3 should report antioxidant activities by risk factors age gender, Tobacco smoking  alcohol consumption, level of activities, and blood prssure weight; calory intake. that will permit to include results of multivariate analysis

Describe in text how it seems that TG and HDL-C have opposite effects on paraoxonase activity  It is quite surprising that the p value  the TG by LDL-C changed from 0887 for double interaction to 0.011 in those with low HDL-C

Consider to add description of the association if any between anti-oxidant activities.

Discussion.

The efforts to support the findings with the literature are interesting but the divergent results highlight the difficulty of the topic and the potential value of adding the findings in a descriptive manner as proposed.

Author Response

Reviewer 1

General

This manuscript is an analysis of Baseline characteristics in a propective study in Mexican subjects the Tlalpan 2010 Study.

Answer: We thank the reviewer for taking the time to carefully review our manuscript and for his/her valuable comments. The number lines in this response correspond to the tracked version of our manuscript.

Particular

Q1. The design paper in BMJ open does not describe this substudy of anti-oxidant activities acccording to risk factors.

A1. Thank you for the observation. The study presented in our manuscript included participants of the ‘Tlalpan 2020 study’ (approval number 13-802); however, it was designed as a sub-study after the Tlalpan 2020 study received ethics approval. We have added the information regarding the ethics approval of this sub-study in lines 87-88: “…while the sub-study presented here was approved under number 15-947.”

Q2. Six antioxidant actitivities have been measured in 626 men and women. The major issue is the focus given to the manuscript with multiple statistical analyses without any adjustment for multiplicity. 15 comparisons in table 2, 35 in table 3 of which  of 3 are statisticant at the p<0.05 level, which could represent simply chance findings. This is threatening the full construct based on double and triple interactions.

A2. We understand the concerns of the reviewer regarding multiplicity, as multiple comparisons could lead to false positives; however, according to the sparsity of effects principle and the hierarchical order principles, when interactions are being analyzed, they are little likely to occur, even more when it comes to triple-interactions (Montgomery, 2013, ISBN 978-1-118-14692-7; Wu and Hamada, 2000 ISBN 978-0-471-25511-6; Box et al., 2005, ISBN 978-0-471-71813-0); for instance, in the work by Bergquist et al. (2011), they found a probability of 0.02 for triple interactions. Also, Montgomery (2013) suggests that an interaction for which a p-value of 0.1 is obtained could be considered as mild.

In addition, the methods to adjust for multiplicity (e.g. Bonferrroni p-value adjustment to control the familywise error rate, the false discovery rate [Benjamini-Hochberg method], and the false acceptance rate) are designed for independent comparisons or hypothesis (Benjamini and Hochberg, 1995; McDonald, 2014). We did not make independent comparisons; the model we used, it would be represented as follows:

Y=a +b1*X1+b2*X2+b3*X3+b12*X1*X2 + b13* X1*X3+ b23* X2*X3+ b123* X1*X2*X3+e

(lines 185-186)

Where X1 = ↑TG, X2 = ↑LDLC, and X3 = ↓HDLC.

So, the variables (↓HDLC, ↑LDLC, and ↑TG) would be conditioned to what happens with the other variable(s) with which they are potentially interacting, losing the independence between comparisons/hypothesis that is assumed for the methods of adjustment for multiplicity. That was one reason for not considering the multiplicity adjustment; however, given the importance that false positives could have, we decided to report both the significances we obtained first and the p-critical value we further obtained by means of the false discovery rate (FDR) procedure by Benjamini-Hochberg (2011); reporting both analysis is suggested by McDonald (McDonald, 2014). To do the adjustment for multiplicity, we considered each term in the linear regression model as an independent hypothesis. We used the Benjamini-Hochberg critical value, P<(i/m)q , where i is the rank that corresponds to the place the comparison has when all the comparisons are ordered from the lowest to the highest p-value, m is the total numbers of tests, and q is the false discovery rate. We considered m=15 for table 3, and m=35 and a false discovery rate of 5% for the results that are now in table 4 and supplementary table 1. We are providing these calculations as ‘Supplementary material (multiplicity adjustment)’ When this adjustment is considered, only the effect of ↑TG on FRAP in the model that only contains main effects would be statistically significant. We stated all this information in lines 418-430:

‘It is worth mentioning that some statistical considerations on this work could be made: we obtained significant interactions in our factorial models, which in practice are very unlikely to occur [25, 48, 49]; thus, if they are observed, there is a good chance that they are true interactions; however, there is now the justifiable trend to carry out adjustments for multiplicity in linear regression models, with the purpose of avoiding false positives. For this reason, we are reporting the calculation of p-critical values by the procedure of Benjamini and Hochberg [50] as ‘Supplementary material (multiplicity adjustment)’, although it should be taken into account that this type of procedures is applied when independent comparisons or hypothesis are being tested [50, 51] and in our case the variables (↓HDLC, ↑LDLC, and ↑TG) are conditioned to each other because of the interaction terms. If the calculated p-critical values were used for our hypothesis testing, then only the effect of ↑TG on FRAP in the model that only contains main effects would be statistically significant. Due to McDonald [51] recommends reporting both the analysis adjusted by multiplicity and the unadjusted one, to avoid the loss of potentially relevant biological findings, they both are reported in this manuscript.’

Q3. The proposal is to fully reorganize the manuscript to describe the measured antoxidant activities and their relationsghips according to lipid levels, their defined subgroups and the risk factors, precisely documented in this prospective cohort. That would largely increase the merit and the interest to readers of the study for readers.

A3. We rearranged our manuscript by beginning our section results with all the descriptive data: table 1, which was slightly modified by adding other variables the reviewer suggested, and table 2 (former table 5). In table 2 the activity of the enzymes is described according to different combinations of alterations in lipid profile (seven groups) and multiple comparisons were made.

We added supplementary information about the antioxidant parameters as a function of the factors we considered for this study (Supplementary table 2). As the reviewer pointed out, this information could be of interest to readers.

We wrote a paragraph in section ‘2.4 Statistical analysis’ (lines 168-176) and two paragraphs in ‘3.1. General characteristics of participants’ (lines 216-231) to mention these additional analyses.

Q4. The title should state that the selected population does not have cardiovascular disease and types of dyslipidemia replaced by lipid profile.

A4. Thank you for your suggestion, we changed the title of our manuscript to: “Activity of antioxidant enzymes and their association with lipid profile in Mexican people without cardiovascular disease: an analysis of interactions”.

Abstract

Q5. Line 30 state separately; line 33 state ceruleoplasmin; rewrite the main finding and in the conclusion perplace mechanistic by prospective or longitudinal studies.

A5.a) The word “separate” is no longer more in the abstract, because this section was rewritten.

b) After searching for the correct spelling of “ceruloplasmin”, we found that the word can be written as “ceruleoplasmin”, “caeruloplasmin” and “ceruloplasmin”, being the first de the less and the later the most common forms; thus, we decided to write it as ceruloplasmin (lines 29 and 40).

                c) We have rewritten the main findings and the conclusion (lines 35-43)

                b) We replaced “mechanistic studies” by “longitudinal studies” (line 43)

Introduction

Q6. Rewrite line 59-60 has this manuscript does not report events but their risk

A6. We changed the sentence “it is not known whether the association between dyslipidemia and the activity of antioxidant enzymes exists before a cardiovascular event occurs” to “…activity of antioxidant enzymes in a context free of cardiovascular disease” (lines 67-68).

Q7. 2.1 subjects and risk factors are fine; consider however to add blood pressure anf weight figures

A7. We added blood pressure, weight, and body mass index as well in table 1, and added the respective information in section 2.1. to describe the methods used to obtain such information (lines 112-118).

Q8. 2.2 please move here the classification given line 171-174

A8. Done. The classification is now in section 2.2.

Q9. 2.3  antioxidant activities explain how SOD1  and catalase activities were reported in U/mg or ?kappa/mg protein

A9. Those units are the standard way to report the activity of those enzymes (10.1038/nprot.2009.197)

Q10. 2.4 statistical analysis. this is the main isuue with proposal to abandon most of statistical analysis except those associated with the lasrt 2 rows of  table 4 on paraoxonase.. line 146 what is meant by considering relevant; line 152 which log transformation were done

A10. The observations of the reviewer lead us to consider to provide additional information regarding one of the main advantages of the analysis we used: “due to the strategy of studying interactions has the advantage that it allows observing how a factor may influence the effect of another, something that is not possible when each factor is evaluated separately [25]” (lines 308-310)

We modified table 4 (former table 3) keeping only the results for paraoxonase 1 and ceruloplasmin activities, for which statistical significance was obtained, while the rest of results of this analysis are now in supplementary table 1. We cannot just consider the terms in the last two rows for PON1, due to in this kind of factorial analysis all the terms should be used (Montgomery, 2013, ISBN 978-1-118-14692-7), that is to say the terms of main effects (↓HDLC, ↑LDLC, and ↑TG for our case), and all the interaction terms (double and triple interactions for our case) that are part of the analysis.

We consider that completely abandon the analysis could lead to the loss of valuable information: the potential interactions among components of the lipid profile that could explain the contradictory results in other studies (those mentioned in the discussion section); however, we think that the suggestion of the reviewer in Q3 of providing descriptive data increases the value of this paper. For this reason, we added the information and did the modifications mentioned in our answers A2 and A3, instead of just leaving aside our previous analysis.

Results

Q11. Table 1 please detete the stars which are not necessary.

A11. Stars from table 1 were deleted.

Q12. Table 2 should be table 5 with the same order for parameters than the one in the method section

A12. Done. Former table 5 is now table 2.

Q13. Table 3 should report antioxidant activities by risk factors age gender, Tobacco smoking  alcohol consumption, level of activities, and blood prssure weight; calory intake. that will permit to include results of multivariate analysis

A13. We included the Supplementary table 2 with associations/comparisons between the activity of antioxidant enzymes and the other variables included in our study. For the reasons exposed in A9, we kept table 3 (previously table 2) as it was in the version of the manuscript you kindly reviewed.

Q14. Describe in text how it seems that TG and HDL-C have opposite effects on paraoxonase activity  It is quite surprising that the p value  the TG by LDL-C changed from 0887 for double interaction to 0.011 in those with low HDL-C

A14. As stated in lines 268-272, the results presented in table 5 (former table 4) are a sub-analysis that derives from the findings in table 4 (former table 3): “To detect which was (were) the double interaction (s) and what was the third variable, the analysis was split first into LDLC and normal LDLC to test the TG-HDLC double interaction; then, into TG and normal TG to test the LDLC-HDLC double interaction; and, finally into ↓HDLC and normal HDLC to test the LDLC-TG double interaction. Results from this sub-analysis are presented in table 5.”; thus, is not that p-value changes from 0.887 to 0.011 but that the p-value of 0.887 (non-significant) is for the interaction only between ↑TG and ↑LDLC without considering the effect of ↓HDLC. Then, when the sub-analysis of the triple interaction was made (table 5), it becomes evident that an interaction between ↑TG and ↑LDLC only occurs when HDLC is low. We clarify this in lines 325-330:  “One would expect that when HDLC is low, the activity of PON1 should be low; however, the sub-analysis of the triple interaction (table 5) showed that the interaction between LDLC with TG, that was not significant in table 4 (p=0.887) when the effect of HDLC was not considered, becomes significant (p=0.011) only when HDLC was low. Thus, under those very specific conditions, the combination of LDLC with TG was associated with a relatively high level of activity of PON1 that is not in accordance with the presence of HDLC.”

Q15. Consider to add description of the association if any between anti-oxidant activities.

A15. We performed a supplementary analysis (Supplementary table 3) describing the associations between the antioxidant parameters we studied. We added a paragraph in this regard in lines414-417: “In addition, we reported some associations between the antioxidant parameters in our study (supplementary table 3) that show that most of them are interrelated, being SOD1 activity the only one that was correlated with all the other parameters; these associations need to be further studied.”

Discussion

Q16. The efforts to support the findings with the literature are interesting but the divergent results highlight the difficulty of the topic and the potential value of adding the findings in a descriptive manner as proposed.

A16. We understand that interactions have their degree of complexity in terms of their interpretation and that was the main reason to include the sub-analysis of the triple interaction for PON1.

We have included descriptive information that the reviewer kindly suggested us in the previous comments and have modified the discussion accordingly (lines 381-382, 404, and 463-466).

Reviewer 2 Report

Dear authors, 

Congratulations on a very well written paper which clearly states the methods and provides a robust and balanced discussion of the results. Please see below two very minor comments.

Consider the inclusion of a figure to describe the exclusion criteria for volunteers and final number of 626.

Expand the limitations sections to be fully comprehensive of the potential biases of the participants. 

Author Response

Reviewer 2

General

Dear authors,

Congratulations on a very well written paper which clearly states the methods and provides a robust and balanced discussion of the results. Please see below two very minor comments.

Answer: We thank the reviewer for his/her comments and his/her time to review our manuscript.}

Particular

Q1. Consider the inclusion of a figure to describe the exclusion criteria for volunteers and final number of 626.

A1. Thanks for your suggestion. A flow diagram of participant was included (Figure 1) after the first paragraph of section 2.1.

Q2. Expand the limitations sections to be fully comprehensive of the potential biases of the participants.

A2. We have stated the potential biases of our study as follows: “In addition, we identified two potential biases of this study:1) the participants do not constitute a random sample, as they voluntarily attend to be evaluated at the INCICh, limiting the generalizability of the results, and 2) data on physical activity, dietary intake, alcohol consumption, and smoking habit were self-reported by the volunteers, so they are subject to a memory bias.” Lines (438-442).

Round 2

Reviewer 1 Report

I think that the authors have taken comments in consideration in the revised manuscript; in particular they have move the actual antioxidant concentrations prior to their statistical evaluation of interactions which remains in one way new but limited in terms of implications since differences were minimal  in actual concentrations of antioxidants. They have also strengthen the description of this substudy as part of their main project, added demographic characteristics and limitations.